# Acceptability of Pharmacogenetic Testing among French Psychiatrists, a National Survey

**DOI:** 10.3390/jpm11060446

**Published:** 2021-05-21

**Authors:** Benjamin Laplace, Benjamin Calvet, Aurelie Lacroix, Stephane Mouchabac, Nicolas Picard, Murielle Girard, Eric Charles

**Affiliations:** 1Esquirol Hospital Center, Department of Research and Innovation, 87000 Limoges, France; benjamin.calvet@orange.fr (B.C.); aurelie.lacroix@ch-esquirol-limoges.fr (A.L.); Murielle.girard@ch-esquirol-limoges.fr (M.G.); 2Inserm UMR1094, Tropical Neuroepidemiology, Institute of Neuroepidemiology and Tropical Neurology, Université de Limoges, 87025 Limoges, France; 3Saint-Antoine Hospital Center APHP, Department of Psychiatry, Sorbonne University, 75012 Paris, France; stephane.mouchabac@aphp.fr; 4iCRIN Psychiatry (Infrastructure of Clinical Research in Neurosciences-Psychiatry), Brain and Spine Institute (ICM), Sorbonne University, INSERM, CNRS, 75013 Paris, France; 5Limoges University Hospital Center, Department of Pharmacology, Toxicology and Pharmacovigilance, 2 Avenue Martin-Luther-King, 87042 Limoges, France; nicolas.picard@unilim.fr; 6Department of Pharmacology, Faculty of Pharmacy, University of Limoges, 2 rue du Docteur-Marcland, 87025 Limoges, France; 7Esquirol Hospital Center, Department of Adult Psychiatry, 87000 Limoges, France; echarles87@orange.fr

**Keywords:** pharmacogenetics, psychiatry, acceptability, survey, France

## Abstract

Psychiatric disorder management is based on the prescription of psychotropic drugs. Response to them remains often insufficient and varies from one patient to another. Pharmacogenetics explain part of this variability. Pharmacogenetic testing is likely to optimize the choice of treatment and thus improve patients’ care, even if concerns and limitations persist. This practice of personalized medicine is not very widespread in France. We conducted a national survey to evaluate the acceptability of this tool by psychiatrists and psychiatry residents in France, and to identify factors associated with acceptability and previous use. The analysis included 397 observations. The mean acceptability score was 10.70, on a scale from 4 to 16. Overall acceptability score was considered as low for 3.0% of responders, intermediate for 80.1% and high for 16.9%. After regression, the remaining factors influencing acceptability independently of the others were prescription and training history and theoretical approach. The attitude of our population seems to be rather favorable, however, obvious deficiencies have emerged regarding perceived skills and received training. Concerns about the cost and delays of tests results also emerged. According to our survey, one of the keys to overcoming the barriers encountered in the integration of pharmacogenetics seems to be the improvement of training and the provision of information to practitioners.

## 1. Introduction

Psychiatric disorders affect many people around the world and are associated with a significant burden of illness. For example, depressive disorders are associated with a lifetime prevalence of one in five people and, according to the World Health Organization (WHO), are the leading cause of disability worldwide since 2017 [1]. The pharmacological treatment of these psychiatric disorders is based on the use of psychotropic drugs. Unfortunately, existing psychotropic drugs, including antidepressants, antipsychotics and mood stabilizers, are associated with unsatisfactory response rates [2,3]. They are also often associated with side effects that can be disabling and may decrease adherence [4]. The high inter-individual variability of responses to psychotropic drugs and the low response rates after a first line of treatment raise the problem of predicting the success of the treatment, which will only be adapted later using a trial-and-error strategy.

Among the factors explaining this variability, genetics occupies an important place, due to its influence on the targets involved in the pharmacokinetics (e.g., Cytochrome P450) and pharmacodynamics of psychotropic drugs [5,6,7,8,9]. Pharmacogenetics is the branch of pharmacology that studies the influence of genome variability on drug response. The use of pharmacogenetic testing is widely spread in clinical practice in certain domains, as numerous pharmacogenomic biomarkers have been identified and translated into clinical practice resulting in updates regarding the prescriptions of drugs such as car-bamazepine, warfarin and abacavir [10]. Pharmacogenetic tests allow the analysis of genes coding for metabolic enzymes, transmembrane transporters or receptors. The use of this information could make it possible to promote the practice of personalized medicine by guiding the choice of treatment towards the most appropriate molecule for a given patient [11], as certain studies are beginning to demonstrate in the treatment of depressive disorders with antidepressants [12,13,14,15,16]. Fewer data are available for antipsychotics and mood stabilizers, but the observations are the same [17]. In a recent review, Van Schaik et al. [18] highlighted the increasing use of pharmacogenetics in the clinical practice of psychiatry.

Although this discipline is in constant expansion, and research on this subject is promising, it remains underused in the field of psychiatry, especially in France. Indeed, according to the French Biomedicine Agency, only 945 tests have been performed for psychiatric treatment in 2018 [19]. According to the experts of the Francophone Network of Pharmacogenetics (RNPGx), this low use would be based more on a lack of knowledge of the interest of these tests by prescribing physicians, than on scientific or organizational reasons [20]. Currently, the main indications of pharmacogenetic testing in France are anti-cancer drugs (dihydropyrimidine dehydrogenase testing before 5-fluorouracil prescription), immunosuppressants (thiopurine-methyltransferase testing before azathioprine pre-scription) and abacavir (HLA-B*5701 screening) [21,22,23,24].

Acceptability by psychiatrists is an important factor, which is likely to influence the integration of pharmacogenetic tests into routine practice. Indeed, for a tool to be used by professionals, it is necessary that they intend to use it and perceive it as useful, reliable and with low risk. Considering this low use of pharmacogenetics in French psychiatry, this raises the question of the acceptability of pharmacogenetics by French psychiatrists and, more broadly, of their attitude and perception towards this tool.

Throughout the world, there are also relatively few articles in the literature on this subject among psychiatrists. In 2008, Hoop et al. [25] found that 45 American psychiatrists had a rather favorable attitude to testing but a low level of training. Later, in 2010, a new study [26] confirmed this positive attitude, correlated with the practice of testing, but without a clear consensus concerning the risks. Later studies also found high levels of interest and even satisfaction in American, Canadian and Japanese populations when the practitioner had already prescribed tests [27,28,29]. Some concerns were highlighted in the study conducted by Chan et al. [30] in Singapore, among 194 psychiatrists and pharmacists. Indeed, more than 80% of responders were concerned about unclear recommendations or a cost that was considered important. Similarly, Dunbar et al. [31] found that responders were concerned about the modalities for receiving results or about the risk of relying on test results at the expense of the clinic.

As far as we know, there is no study that has evaluated these parameters at the national level in France. The main objective of the present study was therefore to assess the acceptability of pharmacogenetics and pharmacogenetic testing by psychiatrists and psychiatry residents in France using a four domains acceptability model based on International Organisation for Standardization (ISO) and Nielsen models (usefulness, usability, easiness, and risk) that was already used to assess medical technologies acceptability by psychiatrists in two previous studies by Bourla et al. [32,33]. Our secondary objectives were to determine the factors associated with and influencing the acceptability score in order to elicit the barriers and pathways likely to impact the integration of this tool in psychiatric care, to determine the factors associated with previous use of pharmacogenetic testing, and to assess more broadly the perception and attitude of psychiatrists towards pharmacogenetics and pharmacogenetic testing, as well as the training received in pharmacogenetics and future training wishes.

## 2. Materials and Methods

### 2.1. Target Population and Sample Composition

The target population for this survey was psychiatrists and residents practicing in France, regardless of their location or mode of practice. The target population was thus composed of approximately 15,000 subjects based on the following estimates: approximately 13,000 psychiatrists and nearly 2000 psychiatry residents.

### 2.2. Development of the Questionnaire

#### 2.2.1. Development

The survey’s questions were designed after several working sessions between the principal investigator, research staff from Esquirol Hospital Center, pharmacogenetics specialists from the University Hospital Center of Limoges and external consultants, psychiatrists and sociologists, experienced in conducting similar surveys. These questions were partly inspired by previous surveys, particularly with regard to sociodemographic data [32,33] or the limits and risks of pharmacogenetics [26,30,34]. The questionnaire was specifically developed for this study and has not yet been validated, but the acceptability model has already been validated [32,33]. To ensure its comprehensibility, the questionnaire was also tested on a sample of psychiatrists and residents.

#### 2.2.2. Structure of the Questionnaire and Collected Data

In the first part of the survey, sociodemographic and epidemiological data were collected, including gender, age, year of graduation, status (resident, hospital practitioner, private practitioner, fellow, assistant, professor), department, place of practice (university hospital, psychiatric hospital, general hospital, private practice, other), the practice area (adult psychiatry, child psychiatry, old age psychiatry, addictology, forensic psychiatry, other) and the theoretical approach (neurobiological, integrative, psychoanalytic, cognitive and behavioral, systemic, other) were collected.

In the second part, the acceptability and perception of pharmacogenetics were assessed in the following areas: perceived competence, history of training and prescription in pharmacogenetics, usefulness, intent to use, reliability and ease of use, perceived limits and risks, and future training wishes. Most of the answers were given in the form of 4-point Likert scales, balanced between positive and negative, with no neutral value, evaluating the respondents’ attitude towards the concerned item according to the following answers: “yes definitely”, “rather yes”, “rather no”, “not at all”. A blank field allowed us to collect qualitative data in the form of feedback at the end of the survey. Altogether, the survey consisted of 42 items and filling time was approximately 10 min (Appendix A).

#### 2.2.3. Assessment of Overall Acceptability Score

An overall acceptability score was calculated. In doing so, we relied on the definition of acceptability given by Bourla et al. in their studies on the acceptability of Artificial Intelligence in psychiatry [32] and on the acceptability of repetitive transcranial magnetic stimulation (rTMS) [33]. They define the acceptability of a technology according to a 4-variable model that makes it possible to evaluate the factors that prevent or, on the contrary, encourage their use: usefulness, intent to use, reliability (and ease of use), risks (and limits). For each variable, the mean score was calculated based on the responses given to the items (rated from 1 to 4 using the 4-point Likert scale). This resulted in a composite overall acceptability score ranging from 4 to 16. Low acceptability was defined by a score ranging from 4 to 8, moderate acceptability ranging from 8 to 12 and high acceptability from 12 to 16.

#### 2.2.4. Assessment of Factor Influencing Acceptability

Comparisons between acceptability scores were carried out to identify any association with socio-demographic characteristics, pharmacogenetic training history and previous pharmacogenetic testing prescription history. Then, a stepwise multiple regression model with forward selection was set up in order to identify the factors influencing acceptability and the different sub-scores.

### 2.3. Survey Diffusion and Data Collection

The participants were contacted via several mailing lists (resident associations, psychiatrist associations, medical boards, hospital centers, local medical associations, etc.). Person-to-person distribution was also encouraged. The self-questionnaire was carried out in electronic form using the Limesurvey software (Limesurvey Gmbh, Hamburg, Germany) hosted on the University of Limoges network.

### 2.4. Statistical Analysis

Quantitative variables were described in the form of means and standard deviation, qualitative variables in the form of numbers and percentages. Differences in the distribution of the quantitative variables between groups were assessed with the non-parametric tests of Mann–Whitney or Kruskal–Wallis, and differences in the distribution between groups of the qualitative variables were explored with the Chi-square test. In order to realize an adjusted linear regression, data that were neither ordinal nor binary were adjusted and organized as follows: residents vs. the others for jobs, private practice vs. the others for place of practice, adult psychiatry vs. the others for practice area, psychoanalytic vs. the others for theoretical approach. Stepwise regression analysis with forward selection was set up to identify which factors were influencing acceptability and the different sub-scores independently from the others. The tests were performed using SPSS Statistics software version 22.00 (Statistical Package 127 for Social Sciences) (IBM^®^, Armonk, NY, USA). The significance level was set at 5%, such that differences with a *p*-value < 0.05 were considered to be significant. Based on a target population of 15,000 subjects, sample size calculation indicated that 375 participants were required to achieve 95% statistical power with an alpha risk of 0.05.

## 3. Results

### 3.1. Survey Implementation

The questionnaire was made available online on the Limesurvey platform from 23 June 2020 to 17 November 2020. The questionnaire was opened 566 times for a total of 440 responses, 393 of which were complete. Among the incomplete answers, those not allowing the calculation of the acceptability score were excluded. The analysis of the results was therefore carried out on the basis of 397 responses (Figure 1).

### 3.2. Socio-Demographic Characteristics

The population in our survey was predominantly feminine (60.5%). The average age was 38.67 years (±12.33). Residents represented only 21% of the sample when the graduated practitioners were mainly hospital practitioners (47% of the respondents).

All the socio-demographic characteristics are presented in Table 1.

### 3.3. Main Result: Acceptability of Pharmacogenetic

#### 3.3.1. Overall Acceptability Score

The overall average acceptability score was 10.70 (±1.34), which was an intermediate average score.

The results showed that the level of acceptability was low for 3.0% of the respondents, intermediate for 80.1% and high for 16.9%.

The lowest 25% of scores ranged between 4 and 9.81.

#### 3.3.2. Perceived Usefulness

The perceived usefulness scores were high, with a mean of 3.25 (±0.51) on a scale of 1 to 4.

Almost all respondents (98.3%) thought that a pharmacogenetic test could improve the response to treatment. A vast majority (90.2%) thought that pharmacogenetics could improve treatment tolerability. Appreciating the necessity of therapeutic adjustment and saving time in treatment appeared to be possible thanks to pharmacogenetics for 89.4% and 82.9% of responders respectively.

The majority (71.8%) of respondents thought that the use of pharmacogenetics was likely to become a common practice in psychiatry.

#### 3.3.3. Intent to Use

A mean of 2.83 (±0.66) on a scale of 1 to 4 was found for the intent to use scores.

A majority of respondents (67.8%) would not prescribe a pharmacogenetic test in the management of all depressive episodes. Intent to use was higher in cases of resistant depression (85.1%). Intent to use in other psychiatric disorders is presented in Table 2.

In a patient whose current treatment seems to be effective, the majority of respondents would not change it (84.6%), of which 28.0% would not change it with certainty.

Also, almost all respondents (93.7%) would accept their doctor prescribing a pharmacogenetic test for themselves.

#### 3.3.4. Reliability and Ease of Use

Reliability and ease of use scores were the lowest of the sub-score, with a mean of 2.05 (±0.39) on a scale of 1 to 4.

47.3% of subjects answered “rather no” to the question on the ease of use of pharmacogenetic tests. 51.4% did not think that pharmacogenetic tests were easy to use. Regarding ease of access, most respondents (90.4%) thought that pharmacogenetic tests were not easy to access. Similarly, the professional recommendations regarding the use of pharmacogenetic tests were mostly considered to be rather unclear or not at all clear (94.5%).

When asked whether doctors’ training and overall level of knowledge of pharmacogenetics was sufficient, a majority of subjects replied “not at all” (61.5%). Only 3.8% of the subjects thought that the level of knowledge could be sufficient.

There were 71.5% of positive responses regarding the reliability of the tests.

#### 3.3.5. Perceived Limits and Risks

The scores for perceived limits and risks were intermediate, with a mean of 2.59 (±0.39) on a scale of 1 to 4.

The perceived limits and risks that most preoccupied the respondents were the cost of the tests (86.6%) and the delays before obtaining the results (64.7%).

Respondents were concerned about the risk of genetic data misuse, incidental discovery of genetic disease, induction of psychological distress in the patient or negative impact on the therapeutic relationship in 35.8%, 41.8%, 24.2% and 8.1% of cases respectively.

Finally, 238 subjects (59.9%) who replied to the survey thought that the benefit–risk balance was generally in favor of carrying out a pharmacogenetic test in the treatment of depression.

The presence of other concerns was expressed by 33 respondents (8.3%). Some mentioned a risk of “dehumanization of the therapeutic relationship and of psychiatric care” or “denial of the human factor”. Others feared the “loss of the clinic” in case of massive use. Others seemed to regret “a certain mercantilism around the tests when the scientific foundations and recommendations deserve to be refined”. The risks linked to the sensitivity of genetic data were specified by some, regarding their “conservation” but also regarding “insurance abuses”. Several of them reaffirmed their “ignorance” of the subject or their impression of a lack of training but also of “public information”. The geographical limit to the accessibility of the tests was also mentioned (“in the town or in the countryside”).

### 3.4. Associated Factors

#### 3.4.1. Prescription History and Perceived Competence

Only 17.1% of responders had previous experience in prescribing pharmacogenetic testing.

Prescription history was significantly associated with job status (*p* = 0.003), place of practice (*p* = 0.008), theoretical orientation (*p* = 0.024), and training history (*p* < 0.001). Among academics (professors and assistants) who responded, 37.5% had already prescribed pharmacogenetic testing, compared with only 7.4% of the private practitioner population. Nearly one third of practitioners working in university hospitals had already prescribed pharmacogenetic testing, compared to 8.0% of those working in the private sector. Previous prescriptions were more frequent among practitioners with a neurobiological approach (27.2%) than among those with a psychoanalytical approach (3.1%).

There was no association with gender, graduation year or practice area (*p* > 0.05) (Table 3).

Responses regarding perceived competences are presented in Table 4.

#### 3.4.2. Training

Less than half of the respondents stated that they had already received some training related to pharmacogenetics (*n* = 182, i.e., 45.8%).

Most of the respondents considered that they had not received sufficient information and training on pharmacogenetics (*n* = 364 or 92.6%). There were 97.2% (*n* = 382) of respondents who were wishing to learn more about pharmacogenetics.

The desired training modalities are presented in Figure 2.

The “other” methods mentioned included “conferences” and “congresses”, with the possibility of holding “practical workshops”, “seminars”, “continuing professional development”, the introduction of “recommendations of good practice” and the writing of “reference books”.

#### 3.4.3. Factors Associated with the Acceptability Score

The distribution of the acceptability score was significantly associated with gender (*p* < 0.04), theoretical approach (*p* < 0.001) and place of practice (*p* < 0.023).

The acceptability score was higher in male responders compared to female, in responders with a neurobiological or cognitive-behavioral approach compared to those with a psychoanalytic or systemic approach. It was also higher among responders practicing in a university hospital center compared to in a private practice (mean acceptability scores are presented in Figure 3).

Training history (*p* < 0.001) and prescription history (*p* < 0.001) were also significantly associated with the acceptability score. A higher score was found in those who had already been trained, as well as in those who had already prescribed. Mean acceptability scores are presented in Figure 4.

Job status, practice area and graduation year were not associated with a difference in the distribution of the acceptability score (*p* > 0.05).

#### 3.4.4. Factors Influencing the Acceptability Score after Linear Regression

The results of the adjusted linear regression after proper data organization are shown in Table 5. The acceptability score was significatively lower for psychiatrists who did not have prior prescription or training history and for those having a psychoanalytic approach.

After injecting the variables into a stepwise linear regression, the training history, alone or in combination with the theoretical approach, or those two previous variables in combination with the prescription history, were influencing the overall acceptability score (Table 6). Gender and place of practice were not significantly influencing the acceptability score according to this statistical model (*p* > 0.05).

Regarding the sub-scores, training and prescription history influenced each of them except for the intent to use, which was only influenced by the theoretical approach with a lower score for psychoanalytic approach (β = −0.118; *p* = 0.030). For limit and risk, an influence of the job status was also found with a lower score for residents (β = −0.166; *p* = 0.008).

## 4. Discussion

### 4.1. Findings

This survey assessed the acceptability of pharmacogenetics and pharmacogenetic testing among psychiatrists and psychiatry residents in France. This is the first study assessing this aspect of pharmacogenetics in France. It revealed an “intermediate” average level of acceptability. The level of acceptability was low for only 3.0% of respondents, intermediate for 80.1% and high for 16.9%. The attitude towards pharmacogenetics and its applications in psychiatry in France seems rather favorable and positive, despite the current very limited use of it. Nevertheless, there are some concerns with persistent preoccupations and the obvious lack of knowledge and training of practitioners. The main factors influencing acceptability were pharmacogenetic training history, theoretical approach and pharmacogenetic testing prescription history.

The different sub-scores used to calculate acceptability were quite heterogeneous. Perceived usefulness obtained the highest sub-score, and most respondents figured that pharmacogenetics would become a common practice in the coming years, as in previous foreign studies [28].

The intent to use seemed to be different according to the proposed disorders. The results are rather positive for resistant disorders, but not in each proposed clinical scenario. This highlights the problem of the proven usefulness of pharmacogenetic testing depending on the disorder presented by the patient. Indeed, pharmacogenetic tests have not been studied for all psychiatric pathologies and are not recommended in all the situations mentioned in the questionnaire. These heterogeneous responses also reflect the need to carry out randomized controlled trials in order to assess the usefulness of the tests and their relevance for each pathology and in possible subgroups of patients [35]. Otherwise, this sub-score was the only one not influenced by training and prescription history. We also wanted to see if there was a risk that practitioners rely too much on the test. We found comparable results to those of McMichael et al.’s study [34], in which only 16% of practitioners would have modified the treatment according to pharmacogenetic recommendations in patients already clinically stabilized by their current treatment.

Reliability and ease of use received the lowest score. Beyond the questions about accessibility or ease of use that intensely concerned nonhospital practitioners, the issue of recommendations considered as unclear remained. In France, there are recommendations for good practice in medical genetics edited by the French Health Authority (HAS) [36], but these are very general and not particularly related to psychiatry. There are also recent recommendations from the RNPGx concerning antidepressants and pharmacogenetics in general [11,21], but no specific recommendations from the HAS or certain societies of psychiatry, such as the French Biological Psychiatry and Neuropsychopharmacology Association (AFPBN), for example [37,38]. This problem seems to exist abroad and throughout the world: despite greater use and more dynamic research in North America, the subject is almost absent from the American Psychiatry Association (APA) and Canadian Network for Mood and Anxiety Treatment (CANMAT) recommendations on the management of mood disorders [39,40]. It is yet worse that the pharmacogenetic guidelines might be slightly different between two different organizations: for example, Clinical Pharmacogenetics Implementation Consortium (CPIC) and Dutch Pharmacogenetics Working Group (DPWG) guidelines diverge for nortriptyline posology adjustment [41].

Also, among the perceived limits and risks, persistent concerns were raised, mainly about the costs of performing the tests and the delays in obtaining the results. Future medico-economic studies will need to be conducted to assess whether, despite the initial cost, the realization of a test is likely to lead to subsequent savings, as already suggested by several studies [42,43]. In our opinion, a parallel can be drawn with the questions about delays of results delivery. While waiting for the results of the test exposes to a potential drug prescription delay of about one or two weeks, a “bad choice” of antidepressant or other psychotropic drug exposes to a risk of losing several weeks or even months in treatment. Moreover, with pre-emptive testing, delays would not be an issue anymore. The other results concerning the risks are consistent with foreign studies among psychiatrists [26,30] and various health workers [44].

Factors associated with the acceptability score are: history of pharmacogenetic test prescription, history of training in pharmacogenetics, gender, theoretical approach and place of practice. It seems logical to us to find prescription history and training history as elements associated with acceptability, since these practitioners are already familiar with them. These results are partly comparable to those found in studies on the acceptability of AI and rTMS in France [32,33]. They also appear to be comparable with those available abroad in psychiatry [26,30]. Regarding the place of practice, proximity to a university hospital leads to proximity to new developments in biomedical research, of which pharmacogenetic studies are a part. Moreover, the fact of practicing in a university hospital is likely to favor access to training. For the theoretical approach, psychoanalysis appears to be less compatible with those more recent, mainly neurobiological, approaches.

By performing a stepwise linear regression, we worked out that the factors really influencing the acceptability were pharmacogenetic training history, theoretical approach and pharmacogenetic testing prescription history. Since changing the theoretical approach of psychiatrists is neither feasible nor honestly desirable, the factors that should be addressed to favor the implementation of pharmacogenetic testing in clinical practice are the training and the possibility of realizing tests.

Unfortunately, the perceived level of competence and received training were found to be low. Given that acceptability is associated with these factors, these results keep on suggesting the need to increase access to quality training in pharmacogenetics and to address the issue of lacking information. These findings are in line with the questions raised by the RNPGx on the causes of the limited use of pharmacogenetics, i.e., a lack of knowledge of the tool and its benefits [20]. For psychiatrists, it is more about a need for information than a strong rejection of new technologies. They are waiting for, and they expect to have, more scientifically validated arguments before they pronounce themselves [32]. That is why information and training are crucial for better acceptability. For example, in the field of HIV, French authors have shown the importance of an optimized communication of pharmacogenetic results between the different parties (researchers, clinicians and patients) to promote better acceptability [45]. Studies [46] had already highlighted this fact in the field of pharmacogenetics in general. If the skills of psychiatrists need to be improved in the field of pharmacogenetics, we also think that they should be supported in the use of this tool. In America, the American Society of Health-System Pharmacists (ASHP) [47] considers that pharmacists have a responsibility to play a leading role in the clinical application of pharmacogenetics. The question of collaboration between clinical psychiatrists and genetic pharmacologists seems to be of paramount importance.

Logically, since they felt there was a lack of training in pharmacogenetics, most practitioners were in favor of receiving more training in pharmacogenetics (97.3%). The most cited desired training modalities were university training and E-Learning. Teaching pharmacogenetics initially or in the course of continuing medical education will improve practitioners’ sense of competence and knowledge in pharmacogenetics. This is likely to lead to greater acceptability, which will encourage practitioners to use the tool. However, this remains conditioned by a clarification of the recommendations, conditions of use and a reduction in the risks perceived by practitioners.

The profile of the responders who had already prescribed pharmacogenetic testing were consistent with those observations and the obvious need for pharmacogenetic-related training. Indeed, previous training was strongly associated with previous pharmacogenetic testing prescription.

### 4.2. Limitations

Firstly, it is necessary to keep in mind that this type of study allows associations to be highlighted, but does not enable a causal relationship to be deduced.

Secondly, we can assume the existence of a selection bias due to the methodology of recruiting respondents: we used an online survey, which may increase the risk of recruiting psychiatrists already aware of the new technologies or interested in the subject. Nevertheless, this potential lack of representativeness did not appear to be an issue regarding the assessment of the factors influencing the acceptability, and we still found a wide range of responses.

The next limitation is that some subjects were not addressed by the questions of the survey. Practitioners were not questioned on their position regarding the testing methods (monogenic tests, polygenic tests, openness to pre-emptive tests) or on the exact place of the test in the care dynamic. Similarly, expected information after ordering a test and the level of satisfaction for those who have already prescribed pharmacogenetic testing were not evaluated, but could be the focus of future studies.

Finally, this is a cross-sectional study, so it may be useful to repeat the measures later, to see if opinions change.

## 5. Conclusions

This survey shows a rather favorable acceptability, but confirms the need to improve the training and development of health professionals’ knowledge of pharmacogenetics, particularly in the field of psychiatry. It will be necessary, in order to encourage appropriate and most relevant use, to think about ways of providing this training throughout medical studies, but also after the residency in the context of continuing professional development.

In addition to the need for training, there is also a need for information. Information campaigns could be carried out by certain actors in the field of psychiatry or through the RNPGx. The strengthening of collaboration between clinical psychiatrists and pharmacologists is also strongly encouraged.

Finally, future studies should not only focus on how to better the tests results but also on patients’ attitude towards pharmacogenetics, since they are the first concerned, and this has not yet been assessed in France yet.

## Figures and Tables

**Figure 1 jpm-11-00446-f001:**
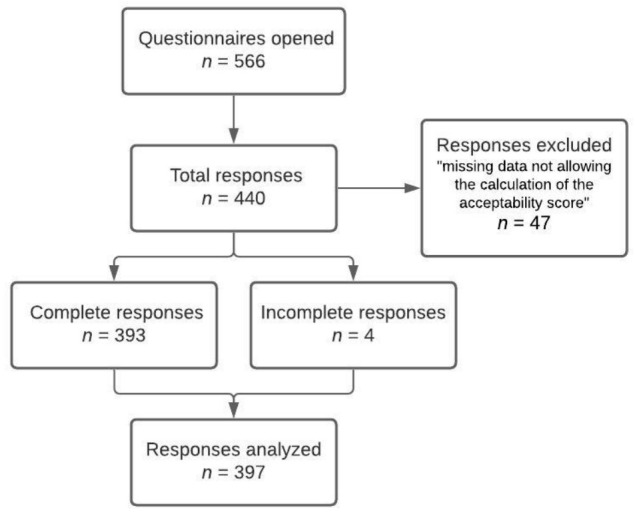
Flowchart of the study.

**Figure 2 jpm-11-00446-f002:**
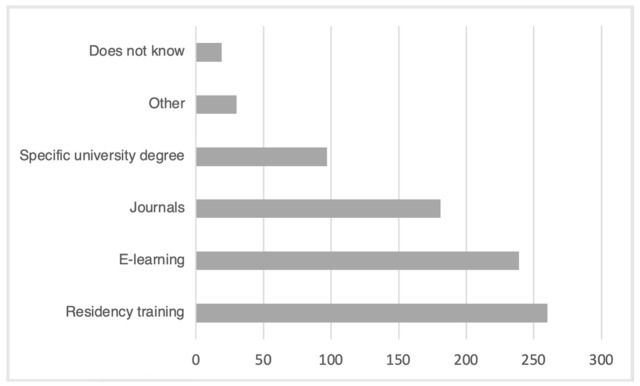
Desired pharmacogenetic training modalities.

**Figure 3 jpm-11-00446-f003:**
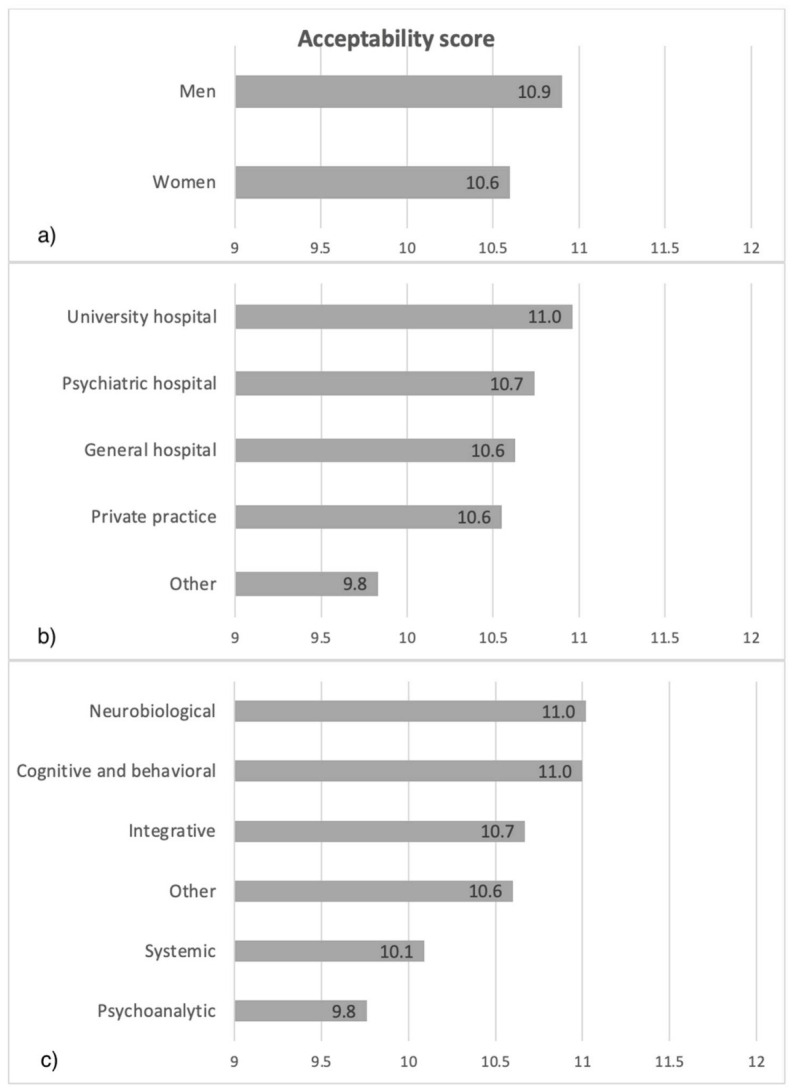
Mean acceptability scores according to gender (**a**), place of practice (**b**) and theoretical approach (**c**).

**Figure 4 jpm-11-00446-f004:**
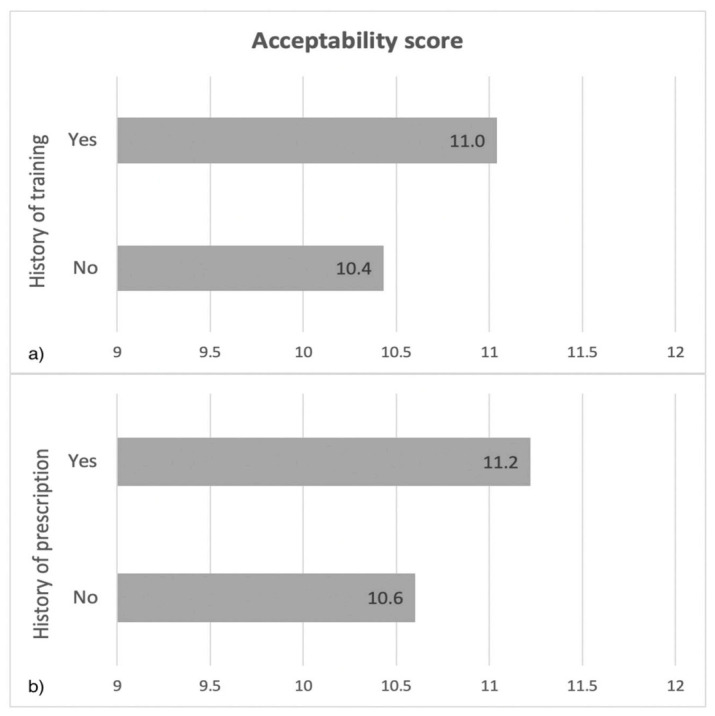
Mean acceptability scores according to history of training (**a**) and prescription (**b**).

**Table 1 jpm-11-00446-t001:** Sociodemographic characteristics of responders.

Variable		*n*	%
Gender	Men	157	39.5%
	Women	240	60.5%
Graduation year	After 2020	106	26.7%
	Between 2016 and 2019	99	24.9%
	Between 2010 and 2015	66	16.6%
	Between 2000 and 2009	56	14.1%
	Between 1990 and 1999	41	10.3%
	Between 1980 and 1989	24	6.0%
	Before 1980	5	1.3%
Job status	Resident	116	29.2%
	Hospital practitioner	187	47.1%
	Private practitioner	54	13.6%
	Assistant	35	8.8%
	Professor	5	1.3%
Place of practice	University hospital center	79	19.9%
	Psychiatric hospital	220	55.4%
	General hospital	26	6.5%
	Private practice	50	12.6%
	Other	22	5.5%
Practice area	Adult psychiatry	274	69.0%
	Child and adolescent psychiatry	83	20.9%
	Old age psychiatry	10	2.5%
	Addictology	19	4.8%
	Forensic psychiatry	1	0.3%
	Other	10	2.5%
Theoretical orientation	Neurobiological	99	24.9%
	Integrative	161	40.6%
	Psychoanalytic	32	8.1%
	Cognitive and behavioral	65	16.4%
	Systemic	23	5.8%
	Other	17	4.3%

**Table 2 jpm-11-00446-t002:** Intent to use according to different psychiatric disorders.

	Answer	Not at All	Rather No	Rather Yes	Yes Definitely
		*n*	*n*	*n*	*n*
Question		%	%	%	%
Would you prescribe pharmacogenetic testing to guide your therapeutic decision in:					
- managing treatment of resistant depression?		29	30	146	192
		7.3%	7.6%	36.8%	48.4%
- managing any depressive disorder?		69	200	86	42
		17.4%	50.4%	21.7%	10.6%
- managing bipolar disorder?		37	127	144	89
		9.3%	32.0%	36.3%	22.4%
- managing treatment of resistant schizophrenia?		27	23	133	214
		6.8%	5.8%	33.5%	53.9%
- managing any schizophrenia?		45	182	109	61
		11.3%	45.8%	27.5%	15.4%
- managing other psychiatric disorders?		41	186	124	46
		10.3%	46.9%	31.2%	11.6%

**Table 3 jpm-11-00446-t003:** Comparison of pharmacogenetic testing history.

		Previous Pharmacogenetic Testing Use?		
		Yes	No	*p* Value
		%	%	
Gender				*p* = 0.053
	Male	21.7	78.3	
	Female	14.2	85.8	
Graduation year				*p* = 0.468
	After 2020	16.0	84.0	
	Between 2016 and 2019	23.2	76.8	
	Between 2010 and 2015	16.7	83.3	
	Between 2000 and 2009	14.3	85.7	
	Between 1990 and 1999	9.8	90.2	
	Before 1989	17.2	82.8	
Job status				***p* = 0.003 ***
	Resident	18.1	81.9	
	Hospital practitioner	15.0	85.0	
	Private practitioner	7.4	92.6	
	Assistant	37.1	62.9	
	Professor	40.0	60.0	
Place of practice				***p* = 0.008 ***
	University hospital center	30.4	69.6	
	Psychiatric hospital	15.5	84.5	
	General hospital	11.5	88.5	
	Private practice	8.0	92.0	
	Other	13.6	86.4	
Practice area				*p* = 0.174
	Adult psychiatry	17.5	82.5	
	Child and adolescent psychiatry	14.5	85.5	
	Old age psychiatry	30.0	70.0	
	Addictology	5.3	94.7	
	Other	36.4	63.6	
Theoretical approach				***p* = 0.024 ***
	Neurobiological	27.2	72.8	
	Integrative	13.7	86.3	
	Psychoanalytic	3.1	96.9	
	Cognitive and behavioral	16.9	83.1	
	Systemic	17.4	82.6	
	Other	17.6	82.4	
Previous PGx training				***p* < 0.001 ***
	Yes	25.3	74.7	
	No	10.2	89.8	

* *p*-value marked in bold are significant value from Chi-square test; PGx = Pharmacogenetic.

**Table 4 jpm-11-00446-t004:** Perceived competences.

	Answer	Not at All	Rather No	Rather Yes	Yes Definitely
		*n*	*n*	*n*	*n*
Question		%	%	%	%
Do you think you are informed enough to identify clinical situations in which testing is indicated?		175	154	57	11
		44.1%	38.8%	14.4%	2.8%
Do you think you are informed enough to explain to patients the risks and benefits of testing?		192	122	69	14
		48.4%	30.7%	17.4%	3.5%
Do you think you are able to adjust your therapeutic decision according to testing results?		87	94	119	32
		26.2%	28.3%	35.8%	9.6%

**Table 5 jpm-11-00446-t005:** Linear regression taking the overall acceptability score as the dependent variable.

	Unstandardized β	Standardized β	*p*-Value	CI	
				Lower	Upper
Gender	−0.181	−0.066	0.187	−0.450	0.088
Age	0.001	0.008	0.945	−0.023	0.025
Graduation year	−0.005	−0.006	0.962	−0.198	0.188
Job status (resident vs. others)	−0.118	−0.040	0.524	−0.482	0.246
Place of practice (PP vs. others)	−0.127	−0.033	0.539	−0.534	0.280
Practice area (adult vs. others)	0.106	0.036	0.469	−0.181	0.393
Theoretical approach (PA vs. others)	−0.794	−0.161	**0.002**	−1.299	−0.288
Prescription history	−0.401	−0.113	**0.025**	−0.751	−0.052
Training history	−0.465	−0.173	**0.001**	−0.736	−0.194

Values marked in bold are significant; CI: 95% Confidence Interval, PP: Private practice, PA: Psychoanalytic.

**Table 6 jpm-11-00446-t006:** Stepwise linear regression taking the overall acceptability score as the dependent variable.

	Unstandardized β	Standardized β	*p*-Value	CI	
				Lower	Upper
Step 1					
Training history	−0.610	−0.227	**<0.001**	−0.869	−0.350
Step 2					
Training history	−0.533	−0.198	**<0.001**	−0.792	−0.274
Theoretical approach (PA vs. others)	−0.866	−0.176	**<0.001**	−1.339	−0.392
Step 3					
Training history	−0.473	−0.176	**<0.001**	−0.735	−0.211
Theoretical approach (PA vs. others)	−0.818	−0.166	**0.001**	−1.290	−0.345
Prescription history	−0.431	−0.121	**0.014**	−0.774	−0.087

Value marked in bold are significant; CI: 95% Confidence Interval, PA: Psychoanalytic.

## Data Availability

Survey data could be made available at any moment if required.

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
