# Peer review of "Acceptability of Pharmacogenetic Testing among French Psychiatrists, a National Survey"

_jpm, 2021, doi:10.3390/jpm11060446_

Round 1
Reviewer 1 Report
The authors conducted a national survey to evaluate the acceptability of pharmacogenetics by psychiatrists and psychiatry residents in France, and also to identify factors associated with acceptability and previous use. The present study is the first one assessing this aspect of pharmacogenetics in France. It has demonstrated that the attitude towards pharmacogenetics and its applications in psychiatry in France seems rather favorable and positive and that one of the keys to overcoming the barriers encountered in the integration of pharmacogenetics seems to be the improvement of training and the provision of information to practitioners in order to ensure appropriate use. I think the manuscript includes new and intriguing findings, however the authors should revise it according to the following minor concerns; The authors should describe more in detail on the use of pharmacogenetics in other countries than France, citing relevant literatures. The authors should describe on the use of pharmacogenetics in other medical fields than psychiatry in France, citing relevant literatures.
Author Response
Dear reviewer,
The authors thank you for your detailed work which helped us improving the manuscript. We have been able to incorporate changes to reflect most of the suggestions provided by the reviewers.
Comment 1: The authors should describe more in detail on the use of pharmacogenetics in other countries than France, citing relevant literatures
Response: Regarding your comment about the use of pharmacogenetics in other countries than France, we agree that it could be useful to add some examples both in psychiatry and in other fields. To do so, we added some example of uses throughout the world :
“The use of pharmacogenetic testing is widely spread around the world in clinical practice in certain domains as numerous pharmacogenomic biomarkers have been identified and translated into clinical practice resulting in updates regarding the prescriptions of drugs such as carbamazepine, warfarin and abacavir (cascella et al 2014)”
We also mentioned the recent review of Van Schaik et al 2020 about the clinical usability of pharmacogenetics in psychiatry:
“In a recent review, Van Schaik et al (Van Schaik et al 2020) highlighted the increasing use of pharmacogenetics in the clinical practice of psychiatry.”
Comment 2: The authors should describe on the use of pharmacogenetics in other medical fields than psychiatry in France, citing relevant literatures
Response: Regarding your second comment about the use of pharmacogenetics in other medical fields than psychiatry in France. We find it really relevant as it allows a comparison with the other medical fields. To address this lack, we added a few words and cited recent articles from RNPGx experts :
“Currently, the main indications of pharmacogenetic testing in France are anti-cancer drugs (dihydropyrimidine dehydrogenase testing before 5-fluorouracil prescription), immunosuppressants (thiopurine-methyltransferase testing before azathioprine prescription) and abacavir (HLA-B*5701 screening) (Picard et al., 2017 ; Quaranta et Thomas, 2017 ; Negrini et Becquemont, 2017 ; Woillard et al., 2017).”
We look forward to hearing from you in due time regarding our submission and to respond to any further questions and comments you may have.
Reviewer 2 Report
The present paper is interestingly approaching an actual medical issue and an important future domain in the personalized and precise medicine, i.e., the acceptability of pharmacogenetic testing in the medical practice using electronic surveys - answering an sensible issue of pharmacological treatment non-compliance in the psychiatric disorders - focused on France psychiatric medical community.
- Title
“Acceptability of pharmacogenetic testing among French psychiatrists, a national survey”
Major
- Abstract
- Line 36 – “according to our survey…”
- Please review the English language correctness and character number
- References
- Most of reference actual
- Please check the correctness of the format
- Introduction
- The actual knowledge clearly described, the knowledge gap and objective clear
- Materials and methods
- Methodology description
- sample size calculation, sample selection method and representability - described
- line 122 – paragraph - In the second part to be outlined from the margin
- Questionnaire
- The questionnaire was previously validated? Please specify
- Please specify the inclusion/exclusion criteria
- (Limitations subsection missing)
- Questionnaire may be added - as supplementary data/annex
- line 153 – description of company, city, country of the Limesoftware
- line 167 – IBM – city, country
- please insert percentages in table 1 and 2
- Line 129 - add (company, town and country) - Microsoft Excel 2010. SPSS Statistics software version 22.00 (Statistical Package 127 for Social Sciences).
- Results
- Tables quite relevant and clear but not included in the body of the paper – it would be more relevant to be attached below the corresponding topic in the text.
- Usage of graphs makes data/tables more understandable
- Figure 1, 2, 3 – resolutions low
- Line 172- a tree map of the survey would be easier to follow
- One decimal for the figures is recommended
- Line 395 – please correct the refference
- Please add if any:
- Limitations
- Please add a list of abbreviations
Author Response
Dear reviewer,
Firstly, the authors thank you for your detailed work which helped us improving the manuscript. We have been able to incorporate changes to reflect most of the suggestions provided by the reviewers.
Please see the attachment

Reviewer 3 Report
This is a clear and well-written manuscript, even if, the paper has some low shortcomings in the text. The introduction is appropriate, although is necessary to add details on pharmacogenetics and its use in the clinical practise. The authors must provide a more adequate explanation on pharmacogenetics and in this regard, they could introduce examples of pharmacogenetic tests widely used in clinical practice (eg. HLA-B * 57: 01, VKORC1). In the literature there are numerous papers that focus on these topics (Cascella et al., Antonatos et al., Fjukstad et al.).
In the discussion and conclusion it is important to emphasize the importance of disseminating knowledge on pharmacogenomics in the medical field.
Author Response
Dear reviewer,
Firstly, the authors thank you for your detailed work which helped us improving the manuscript. We have been able to incorporate changes to reflect most of the suggestions provided by the reviewers.
Comment 1: The introduction is appropriate, although is necessary to add details on pharmacogenetics and its use in the clinical practise. The authors must provide a more adequate explanation on pharmacogenetics and in this regard, they could introduce examples of pharmacogenetic tests widely used in clinical practice (eg. HLA-B * 57: 01, VKORC1). In the literature there are numerous papers that focus on these topics (Cascella et al., Antonatos et al., Fjukstad et al.).
Response: Regarding your relevant comment about the introduction and the use of pharmacogenetic, we added some examples of the main uses in France :
“Currently, the main indications of pharmacogenetic testing in France are anti-cancer drugs (dihydropyrimidine dehydrogenase testing before 5-fluorouracil prescription), immunosuppressants (thiopurine-methyltransferase testing before azathioprine prescription) and abacavir (HLA-B*5701 screening) (19–22).”
We cited recent and relevant articles from RNPGx (Picard et al., 2017 ; Quaranta et Thomas, 2017 ; Negrini et Becquemont, 2017 ; Woillard et al., 2017).
We also added some examples of uses and recent research throughout the world using one of the citations you kindly recommended to us:
“The use of pharmacogenetic testing is widely spread around the world in clinical practice in certain domains as numerous pharmacogenomic biomarkers have been identified and translated into clinical practice resulting in updates regarding the prescriptions of drugs such as carbamazepine, warfarin and abacavir” (cascella et al 2014)
Comment 2: In the discussion and conclusion it is important to emphasize the importance of disseminating knowledge on pharmacogenomics in the medical field.
Response: When it comes to the discussion, we totally agree with your remark. Indeed, the importance of disseminating knowledge on pharmacogenomics in the medical field remains one of the keys to favor its use. To emphasize on this point, we added a few words on it and linked it with previous findings from Bourla et al 2018 and Moutel et al 2005:
“For psychiatrists, it's more a need of information than a strong reject of new technologies. They are waiting and expect to have more scientifically validated arguments before they pronounce themselves (Bourla et al 2018). That’s why information and training are crucial for better acceptability. For example, in the field of HIV, French authors have shown the importance of an optimized communication of pharmacogenetic results between the different parties (researchers, clinicians and patients) to promote better acceptability (Moutel et al 2005).”
We look forward to hearing from you in due time regarding our submission and to respond to any further questions and comments you may have.